# Degradation or Restoration? The Temporal-Spatial Evolution of Ecosystem Services and Its Determinants in the Yellow River Basin, China

**Bowen Zhang** [1,2] , **Ying Wang** [1,2,*] , **Jiangfeng Li** [1] **and Liang Zheng** [3]

1    School of Public Administration, China University of Geosciences, Wuhan 430074, China;
     bw.zhang@cug.edu.cn (B.Z.); jfli@cug.edu.cn (J.L.)
2    The Key Laboratory of the Ministry of Natural Resources for Legal Research, Wuhan 430074, China
3    Changjiang Institute of Survey, Planning, Design and Research, Wuhan 430024, China; zl@cug.edu.cn
*    Correspondence: yingwang0610@cug.edu.cn

**Abstract:** Ecosystem services (ESs) are irreplaceable natural resources, and their value is closely related to global change and to human well-being. Research on ecosystem services value (ESV) and its influencing factors can help rationalize ecological regulatory policies, and is especially relevant in such an ecologically significant region as the Yellow River Basin (YRB). In this study, the ecological contribution model was used to measure the contribution of intrinsic land use change to ESV, the bivariate spatial autocorrelation model was applied to investigate the relationship between land use degree and ESV, and the geographical detector model (GDM) and geographically weighted regression (GWR) were applied to reveal the impact of natural and socio-economic factors on ESV. Results showed that: (1) The total ESV increased slightly, but there were notable changes in spatial patterns of ESV in the YRB. (2) Land use changes can directly lead to ESV restoration or degradation, among which, conversion from grassland to forest land and conversion from unused land to grassland are vital for ESV restoration in the YRB, while degradation of grassland is the key factor for ESV deterioration. (3) According to GDM, NDVI is the most influential factor affecting ESV spatial heterogeneity, and the combined effect of multiple factors can exacerbate ESV spatial heterogeneity. (4) GWR reveals that NDVI is always positively correlated with ESV, GDP is mainly positively correlated with ESV, and population density is mainly negatively correlated with ESV, while positive and negative correlation areas for other factors are roughly equal. The findings can provide theoretical support and scientific guidance for ecological regulation in the YRB.

**Keywords:** ecosystem services value (ESV); natural and socio-economic factors; ecological contribution model; geographical detector model (GDM); geographically weighted regression (GWR); Yellow River Basin (YRB)

## 1. Introduction

Ecosystem services (ESs) refer to all benefits obtained by humans from the natural environment [1]. The ecosystem services value (ESV) is a measure of ESs and is an important indicator of ecological health, which includes transfer of ESs into practical applications [2]. To ensure territorial ecological security, adapt to global climate change, and achieve high-quality development, it is essential to monitor and maintain ESV [3]. However, with socio-economic development and population growth, high-intensity human activities have had a huge impact on ecosystems [4], resulting in a slew of global ecosystem deterioration issues such as climate change [5], ozone layer destruction [6], biodiversity loss [7], water pollution [8] and land desertification [9]. With the increasing prominence of environmental issues, there exists an urgent need to investigate the spatiotemporal evolution of regional ESV and its influencing factors, in order to achieve a balance between ecosystems and socio-economic sustainable development.

Many studies have shown that ESV is vulnerable to multiple impacts from natural changes and human activities, with land use and cover change (LUCC) being the most important influencing factor on terrestrial ESV [10–12]. Changes in land use type cause changes in basic ecological elements, hence influencing ESV directly, although the underlying relationships are complex [13]. Statistical analysis, correlation analysis, regression analysis, redundancy analysis, principal component analysis, and other methods have been employed to explore the relationship between LUCC and ESV [14–17]. Wang et al. [18], for example, used a geographically weighted regression (GWR) model to investigate the effect of LUCC on ESV, finding that forest land and grassland had the greatest effect on ESV. Using bivariate spatial autocorrelation, Lei et al. [19] explored the link between land use degree and ESV and discovered a negative correlation. However, most studies have explored only the effect of area of different land use types on ESV, or the effect of land use degree on overall ESV. Using an ecological contribution model of land use change, we attempted to reveal the potential impact on ESV changes of transformation processes of land use type. In addition, we employed a bivariate spatial autocorrelation model to investigate the relationship between degree of land use and the value of each ES, in order to uncover a more nuanced relationship between them.

Furthermore, a number of natural and socio-economic factors have a substantial impact on overall ESV through exerting varied effects on the inherent aspects of ESs [20,21]. For example, Zhang et al. [22] found that an increase in average precipitation leads to an increase in lake and wetland area, which in turn leads to an improvement in regional ESV. Dai et al. [23] showed that when population density exceeds a threshold, there is a risk of ecological undersupply, which has a negative impact on ESV. When studying the effects of natural and socio-economic factors on ESV, it is necessary to consider a wide range of factors and their interactions. The geographical detector model (GDM) is a new statistical method for revealing the impact of multiple influencing factors and their linkages on a geographical phenomenon [24]. It has two major advantages. First, it can identify relationships between a complex set of factors and a wide range of geographical phenomena, without any assumptions or restrictions regarding independent and dependent variables, allowing it to be used without removing multi-collinear factors [24–26]. Second, it can quantitatively extract the implicit interrelationships between pairs of factors and obtain useful findings [27]. The GDM is now widely utilized in a variety of disciplines, and an increasing number of researchers have employed it to investigate the factors that influence ESV [28,29]. In this study, the GDM's Factor Detector and Interaction Detector tools were used to reveal the relative roles of the multiple natural and socio-economic factors, as well as their interactions. However, the GDM can only quantify the effect magnitude of various factors, and the directions of influence could not be determined [30]. To investigate the direction and spatial variation of each factor's effect on ESV, we further adopted the GWR, which can capture the correlation between spatial objects themselves, as well as reflect the spatial heterogeneity and direction of influence at different geographical locations through the regression coefficient [31].

The Yellow River Basin (YRB) is a key ecological barrier and economic belt in China, and plays a critical role in China's socio-economic development and ecological security [32]. In recent years, the Chinese government has placed high priority on ecological conservation and the green development of the YRB, implementing initiatives such as the Three-North Shelterbelt Project, the "Grain-for-Green" and Natural Forest Protection programs, which have begun to bear fruit [33–35]. However, some areas of ecological degradation still exist in the YRB, where the ESs have been severely damaged. For example, the Shaanxi-Gansu-Ningxia region, in the upper and middle of the YRB, suffers from severe soil erosion and a fragile ecological environment [36]. Severe silt deposition frequently generates river overhangs in the lower YRB, flooding is frequent, and ESs are grave danger [37]. To optimize the YRB's ecological structure and promote high-quality development, it is critical to identify the vulnerable ESV areas in the YRB and reveal their influencing factors. Specifically, the objectives of our study are: (1) to map the spatial distribution of ESV and

identify vulnerable areas of ESV in the YRB; (2) to quantify the impact of LUCC on ESV and reveal the extent of that impact; (3) to investigate the impact of natural and socio-economic factors on ESV, and reveal the interactions of multiple factors; and (4) to propose relevant recommendations based on the findings.

## 2. Materials and Methods

### 2.1. Overview of the Study Area

The Yellow River is the second longest river in China, with a total length of 5464 km. It is a major biodiversity gathering region as well as an ecological security barrier within China. This study defines the provinces through which the Yellow River flows, viewing the YRB in a broad sense, based on the Yellow River and physical geographic watersheds, with the provincial administrative regions considered as the units. Since most of Sichuan Province belongs to the Yangtze River Basin, the other eight provinces where the Yellow River flows were used as research areas in this study (Figure 1). The terrain in the YRB is complex. With an average altitude of around 4000 m, the western region is made up of a succession of mountains with permanent snow and glacier landforms. The central region has a loess landform with loose soil and considerable soil erosion, with an average altitude of 1000 m to 2000 m. The Yellow River's alluvial plain makes up the majority of the eastern area. The overall ecological quality in the YRB is poor due to substantial land degradation including soil erosion and desertification. It is critical to evaluate the YRB's ecological condition, identify its ecological weak spots, to provide scientific guidance for ecological protection and spatial management, and build its ecological barrier status.

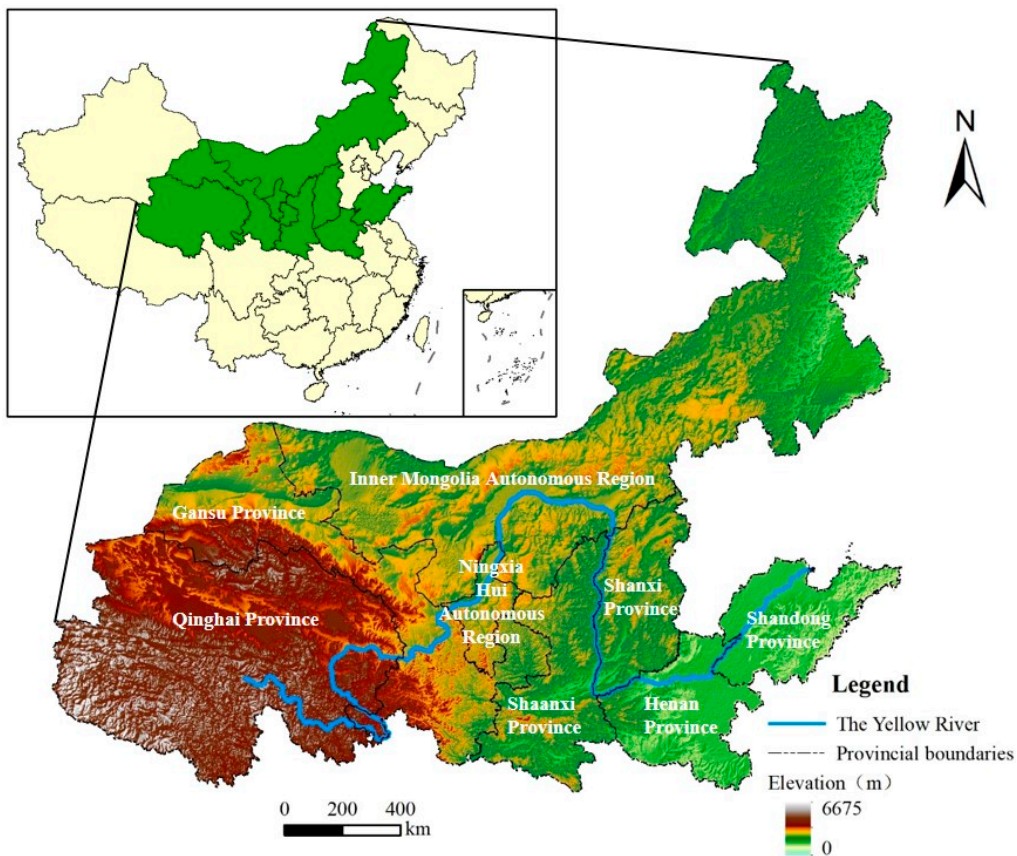

**Figure 1.** The location of the YRB.

### 2.2. Data Sources

This study used basic geographical data, such as LUCC, administrative boundaries, grain yield per unit area and grain price, and data on natural and socio-economic factors of ESV change (Table 1). The LUCC data of 1990, 2000, 2010 and 2018, with a spatial

resolution of 30 m × 30 m, were obtained from the Resources and Environmental Sciences and Data Center, Chinese Academy of Sciences (https://www.resdc.cn/ (accessed on 20 November 2021)). The original LUCC data contains 29 land use types, which we reclassified into 6 categories (i.e., cropland, forest land, grassland, water bodies, and unused land). The administrative boundary data of each administrative unit was taken from the 1:400,000 database of the National Geomatics Center of China (http://www.ngcc.cn/ngcc/ (accessed on 22 November 2021)). Using sown area and grain yield, grain yield per unit area was estimated, with data coming from the statistical yearbook of each region. The grain price data came from the China Agricultural Product Price Survey Yearbook.

**Table 1.** List of basic data information.

| Type | Name | Source | Year | Precision |
|---|---|---|---|---|
| Statistics | Grain sown area Grain yield Grain price | Statistical Yearbook | 2018 | Provincial |
| Vectors | Administrative boundary River map Road map City location | The 1:400,000 database of the National Geomatics Center of China (http://www.ngcc.cn/ngcc/ (accessed on 22 November 2021)) | 2017 | - |
| Rasters | LUCC | The Resources and Environmental Sciences and Data Center, Chinese Academy of Sciences (https://www.resdc.cn/ (accessed on 20 November 2021)) | 1990, 2000, 2010, 2018 | 30 m |
| | DEM | | - | 250 m |
| | Precipitation | | 2015 | |
| | NDVI | | 2018 | |
| | Population density | | | 1 km |
| | GDP | | 2015 | |

The natural factors included elevation, slope, aspect, soil types, soil erosion, precipitation, temperature, vegetation types, and NDVI. Socio-economic factors included population density, GDP, road maps, river maps, railway maps, county location, city location, and provincial capital location. The DEM was processed in ArcGIS 10.7 to produce the elevation, slope, and aspect maps. The Euclidean Distance Tool of ArcGIS 10.7 was used to create the distance maps. Data for other influencing factors were obtained from the Resources and Environmental Sciences and Data Center, Chinese Academy of Sciences. Finally, using the ArcGIS 10.7 software, all the data were converted to raster data with a resolution of 1000 m × 1000 m, and the influencing factors data were discretized to type data sets according to Jenks (Figure 2). In addition, the mapping and tabulation were processed at the prefecture-level city scale.

### 2.3. Methods

#### 2.3.1. LUCC Evolution Analysis Model

(1)  Land use transfer matrix

The transfer matrix of land use can depict the changes in various land use types through time and the amount of change from one land use type to another [38]. It is based on a grid-by-grid description of the change from the initial state to the final state, reflecting the transformation of land use from moment T to moment T + 1, which can reveal the spatial and temporal evolution of land use patterns. The transfer matrix is described as follows:

$$S_{ij} = \begin{bmatrix} s_{11} & s_{12} & \cdots & s_{1n} \\ s_{21} & s_{22} & \cdots & s_{2n} \\ \cdots & \cdots & \cdots & \cdots \\ s_{n1} & s_{n2} & \cdots & s_{nn} \end{bmatrix} \tag{1}$$

where $S_{ij}$ represents the area of LUCC change from type $i$ to type $j$, and $s_{nn}$ denotes the LUCC type before and after transfer.

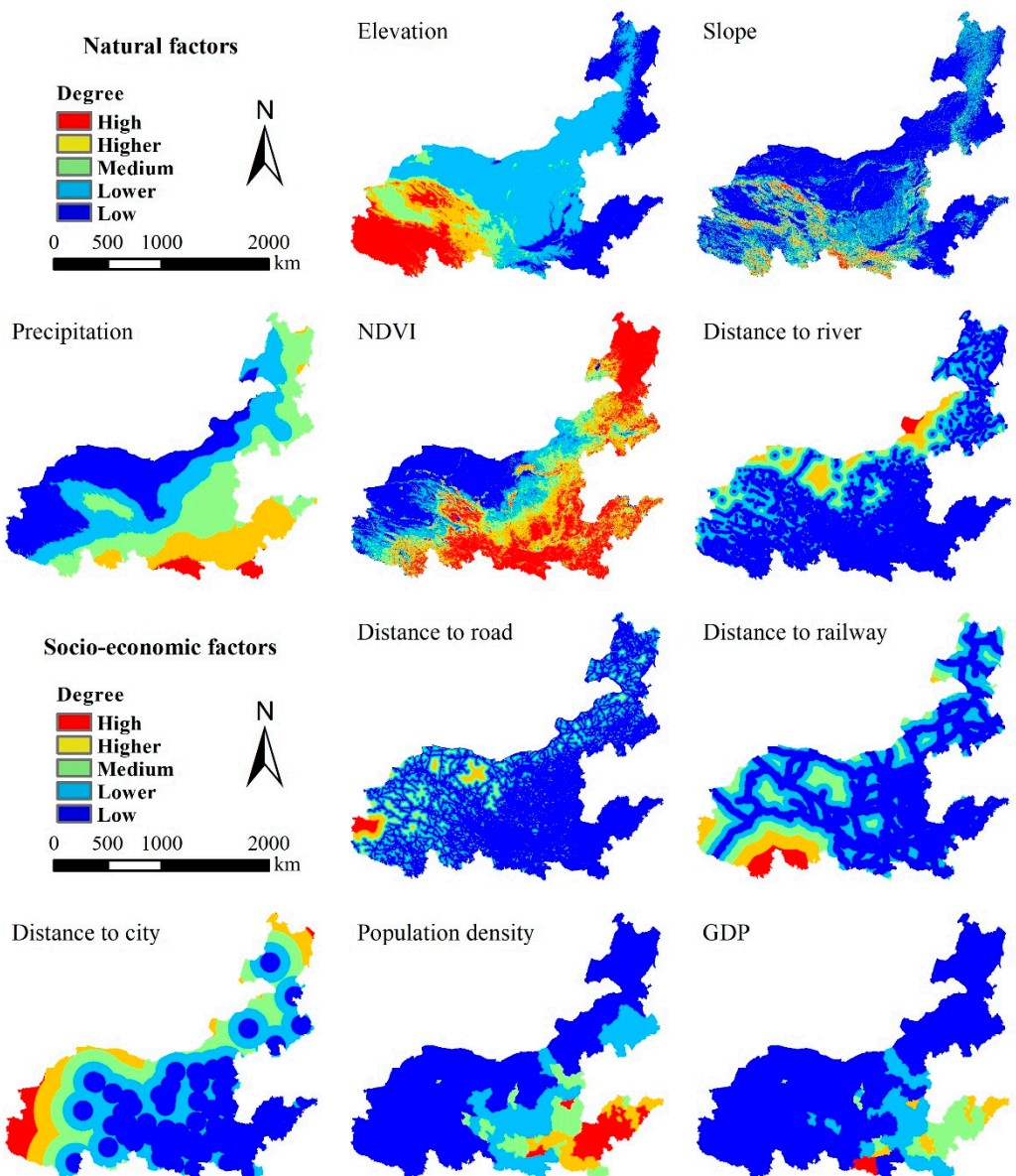

**Figure 2.** Influencing factors of ESV change.

(2)    Calculation of land use degree

The comprehensive index of land use degree is a metric for assessing the extent to which land is used by humans. Different land use types were assigned to distinct values to represent the level of human utilization [39]. Specifically, built-up land was graded 4, cropland graded 3, forest land, grassland and water bodies were graded 2, and unused land was graded 1. The formula is as follows:

$$L = \sum_{i=1}^{n} A_i C_i \qquad (2)$$

where $L$ is the comprehensive index of land use degree; $A_i$ is the grade of different land use type, and $C_i$ is the proportion of land use type $i$ to the total area.

#### 2.3.2. ESV Evaluation Model

To evaluate ESV, this study adopted the value coefficient method modified by Xie et al. [40]. Specifically, the economic value of food supply provided by cropland was defined as the standard value, and the ESV of all other land use types was converted into equivalent values corresponding to the standard value. The economic value of food supply equals to 1/7 of the estimated value of grain yield in the YRB, which can be estimated based on the grain yield per unit area of the YRB and the average grain price in 2018, which is $2.08 \times 10^5$ CNY·km$^{-2}$·a$^{-1}$ (1 USD = 6.70 CNY). The value of each type of ESs provided by different land use types is shown in Table 2. The ESV and its changes in the YRB can be estimated with the following formulas:

$$ESV = \sum_{i=1}^{n}(LUC_i \times VC_i) \tag{3}$$

$$AESV = \frac{\sum_{i=1}^{n}(LUC_i \times VC_i)}{\sum_{i=1}^{n} LUC_i} \tag{4}$$

$$C = \frac{ESV_{t2} - ESV_{t1}}{ESV_{t1}} \times 100\% \tag{5}$$

where $ESV$ is ecosystem services value (CNY), $AESV$ is the average $ESV$ (CNY·km$^{-2}$), $VC_i$ is the $ESV$ coefficient from land use type $i$, and $LUC_i$ is the area of land use type $i$, $C$ is the rate of change of $ESV$, $ESV_{t1}$ and $ESV_{t2}$ represent $ESV$ at $t_1$ and $t_2$, respectively (CNY).

**Table 2.** ESV coefficient of different land use type in the YRB (CNY·km$^{-2}$·a$^{-1}$).

| Ecosystem Services | | Cropland | Forest Land | Grassland | Water Bodies | Unused Land |
|---|---|---|---|---|---|---|
| Supply services | Food supply | 2083.02 | 687.40 | 895.70 | 926.94 | 41.66 |
| | Raw material | 812.38 | 6207.40 | 749.89 | 614.49 | 83.32 |
| Regulation services | Gas regulation | 1499.77 | 8998.65 | 3124.53 | 3041.21 | 124.98 |
| | Climate regulation | 2020.53 | 8477.90 | 3249.51 | 16,257.98 | 270.79 |
| | Hydrological regulation | 1603.92 | 8519.55 | 3166.19 | 33,547.05 | 145.81 |
| | Waste disposal | 2895.40 | 3582.80 | 2749.59 | 30,464.18 | 541.59 |
| Support services | Soil conservation | 3062.04 | 8373.74 | 4665.97 | 2499.62 | 354.11 |
| | Biodiversity maintenance | 2124.68 | 9394.42 | 3895.25 | 7415.55 | 833.21 |
| Cultural services | Aesthetic landscape | 354.11 | 4332.68 | 1812.23 | 9508.99 | 499.93 |
| Total | | 16,455.86 | 58,574.54 | 24,308.85 | 104,276.02 | 2895.40 |

#### 2.3.3. ESV Changes in Response to LUCC

(1)　Ecological contribution model of land use change

To calculate how land use change contributes to ESV change, we used the ecological contribution model of land use change. This method can clearly show the direction and extent of the contribution of different land use changes to ESV change, and facilitate the identification of the main types of land use change that affect ESV [41]. Its formula is as follows:

$$EL_{i-j} = \frac{(VC_j - VC_i) \times LUC_{i-j}}{\sum_{i=1}^{n} \sum_{j=1}^{n} \left[ (VC_j - VC_i) \times LUC_{i-j} \right]} \tag{6}$$

where $EL_{i-j}$ is the contribution of land use change to $ESV$ change, $VC_i$ and $VC_j$ is the coefficient from ESs type $i$ and type $j$, $LUC_{i-j}$ is the total area converted from land use type $i$ to type $j$.

(2)　Bivariate spatial autocorrelation model

LUCC can cause ESV variation. The local bivariate spatial autocorrelation proposed by Anselin [42] was used to investigate the spatial correlation between land use degree and ESV. Its formula is as follows:

$$I_{kl}^i = z_k^i \sum_j w_{ij} z_l^j \tag{7}$$

where $w_{ij}$ is the spatial weight matrix, $X_k^i$ represents the value of attribute $i$ of unit $k$, $X_l^j$ represents the value of attribute $l$ to unit $j$, $\overline{X_k}$ and $\overline{X_l}$ are the average values of attributes $k$ and $l$, respectively, $\sigma_k$ and $\sigma_l$ are the variances of attributes $k$ and $l$, respectively.

### 2.3.4. Geographical Detector Model (GDM)

In this study, the average ESV was taken as the dependent variable, 17 natural and socio-economic factors were taken as independent variables, and the GDM was used to investigate the individual impacts of each factor and their interactions, as well as the degree of impact on spatial heterogeneity of the average ESV in the YRB.

The GDM is composed of Factor Detector, Interaction Detector, Risk Detector and Ecological Detector, which can be used to detect spatial heterogeneity and its influencing factors [24]. In this study, the Factor Detector and Interaction Detector tools were used to explore the impact of natural and socio-economic factors on ESV.

(1)　Factor Detector

The Factor Detector uses the relationship between the within-strata variance and the variance of the entire region to measure the explanatory degrees of independent to dependent variables. The formula is as follows:

$$q = 1 - \frac{1}{N\sigma^2} \sum_{h=1}^{L} N_h \sigma_h^2 \tag{8}$$

where $q$ measures the influence degree of each influencing factor on the dependent variable *ESV*, and its value is within [0, 1]. The larger the $q$ value, the stronger the influence of the factor on *ESV*. $h = 1, 2, \ldots, L$ represents the strata of influencing factors. $N_h$ and $N$ are the number of samples in strata $h$ and the entire region, respectively. $\sigma_h^2$ and $\sigma^2$ are the variance of influencing factors in strata $h$ and the entire region, respectively.

(2)　Interaction Detector

The Interaction Detector is used to quantify the interaction between different factors, i.e., wheter two factors have stronger or weaker effects on ESV when combined than when considered separately. The interaction effects of influencing factors were judged by the relationship between $q(x_i \cap x_j)$, $q(x_i)$, and $q(x_j)$ based on the following formulas:

If $\min(q(x_i), q(x_j)) < q(x_i \cap x_j) < \max(q(x_i), q(x_j))$, it represents single-factor nonlinear weakening.
If $q(x_i \cap x_j) > \max(q(x_i), q(x_j))$, it represents two-factor enhancement.
If $q(x_i \cap x_j) > q(x_i) + q(x_j)$, it represents nonlinear enhancement.
If $q(x_i \cap x_j) = q(x_i) + q(x_j)$, it represents mutual independence.

### 2.3.5. Geographically Weighted Regression (GWR)

By establishing the local regression equation in each grid, GWR can be used to study the correlation between multiple variables with spatial distribution characteristics to a dependent variable. In this study, GWR described the correlation between ESV and natural socio-economic factors, and reflected the spatial heterogeneity and direction of influence through the regression coefficient within each grid [31]. Its formula is as follows:

$$y_i = \beta_o(u_i, v_i) + \sum_{k=1}^{p} \beta_k(u_i, v_i) x_{ik} + \xi_i \tag{9}$$

where: $y_i$ is the ESV in grid $i$, $(u_i, v_i)$ is the space coordinates of grid $i$, $\beta_o$ and $\beta_k$ is the $o$ and $k$ regression coefficient in the grid, $x_{ik}$ is the $k$th independent variable for the $i$th site, $\xi_i$ is the residual value in the grid $i$.

In this study, ten factors that passed the test for multi-collinearity were screened as independent variables and regressed with average ESV as the dependent variable by the GWR model. The results show an adjusted $R^2$ value of 0.82, which indicates that the GWR model fits well for exploration of the ESV and its influencing factors, and the results can be used to explain the spatial heterogeneity of the influencing factors of ESV.

## 3. Results

### 3.1. Characteristics of LUCC Evolution in the YRB

3.1.1. Land Use Dynamics from 1990 to 2018 in the YRB

The land use transfer matrix (Figure 3) shows that the dominant land use types in the YRB are grassland, cropland, and unused land, with these three types accounting for over 80% of the total area during 1990–2018. The percentage of water bodies in the study area is roughly 2%, with a modest increase every year. Built-up land has increased substantially, nearly doubling from 1990 to 2018. Cropland area expanded greatly between 1990 and 2000, then declined, maintaining a marginal overall increase. Overall, the total amount of forest land has fluctuated and increased. Between 1990 and 2010, the amount of grassland declined significantly, especially between 2000 and 2010, when it decreased by 44,325 km². However, since 2010, the downward trend has reversed and its area has gradually increased.

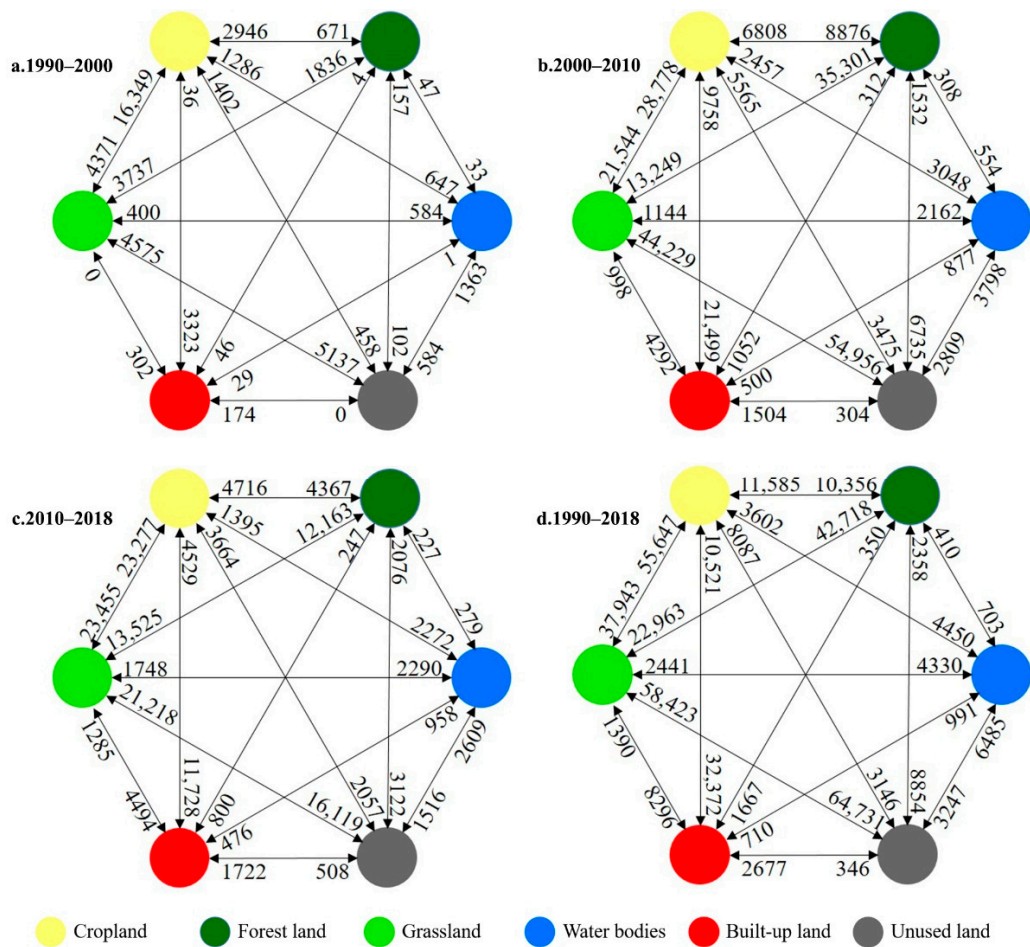

**Figure 3.** Land use transfer matrix in the YRB during 1990–2018 (km²).

3.1.2. Land Use Degree in the YRB

The land use degree of the YRB increased gradually over time, rising from 1.9650 to 1.9861. Meanwhile, more than 90% of the cities studied exhibited a growing trend in land use during the study period. After more than 20 years of development, only cities in the Inner Mongolia

Autonomous Region and along the boundary between eastern Gansu Province and central Shaanxi Province have seen a decline in land use. During 1990–2018, the Alxa League had the lowest degree of land use, at around 1.1275. The cities with the highest degree of land use changed over time, with Zhoukou City (3.1615), Shangqiu City (3.1697), Liaocheng City (3.2066), and again Liaocheng City (3.2105) being the highest in 1990, 2000, 2010, and 2018, respectively. These three cities are located in Henan or Shandong Provinces, near coastal areas with developed agriculture or industry.

*3.2. Temporal and Spatial Variations of ESV in the YRB*

As shown in Figure 4, the lower ESV area in the YRB is concentrated in the northwest, and the high value area is concentrated in the northeast. According to the calculation results, the average ESV was $227.29 \times 10^4$, $226.35 \times 10^4$, $227.22 \times 10^4$, and $227.43 \times 10^4$ CNY·km$^{-2}$ in 1990, 2000, 2010, and 2018, respectively. The regulation services had the highest value, accounting for more than 50% of the total ESV, and that with cultural services it followed a changing pattern of falling and growth over the study period. Eventually, their values change from $356.81 \times 10^{10}$ CNY and $49.03 \times 10^{10}$ CNY to $359.01 \times 10^{10}$ CNY and $49.16 \times 10^{10}$ CNY, respectively. Support services, on the other hand, steadily declined from $211.68 \times 10^{10}$ CNY to $209.77 \times 10^{10}$ CNY. In addition, supply services fluctuated, but their overall value remained consistent at roughly $62.51 \times 10^{10}$ CNY.

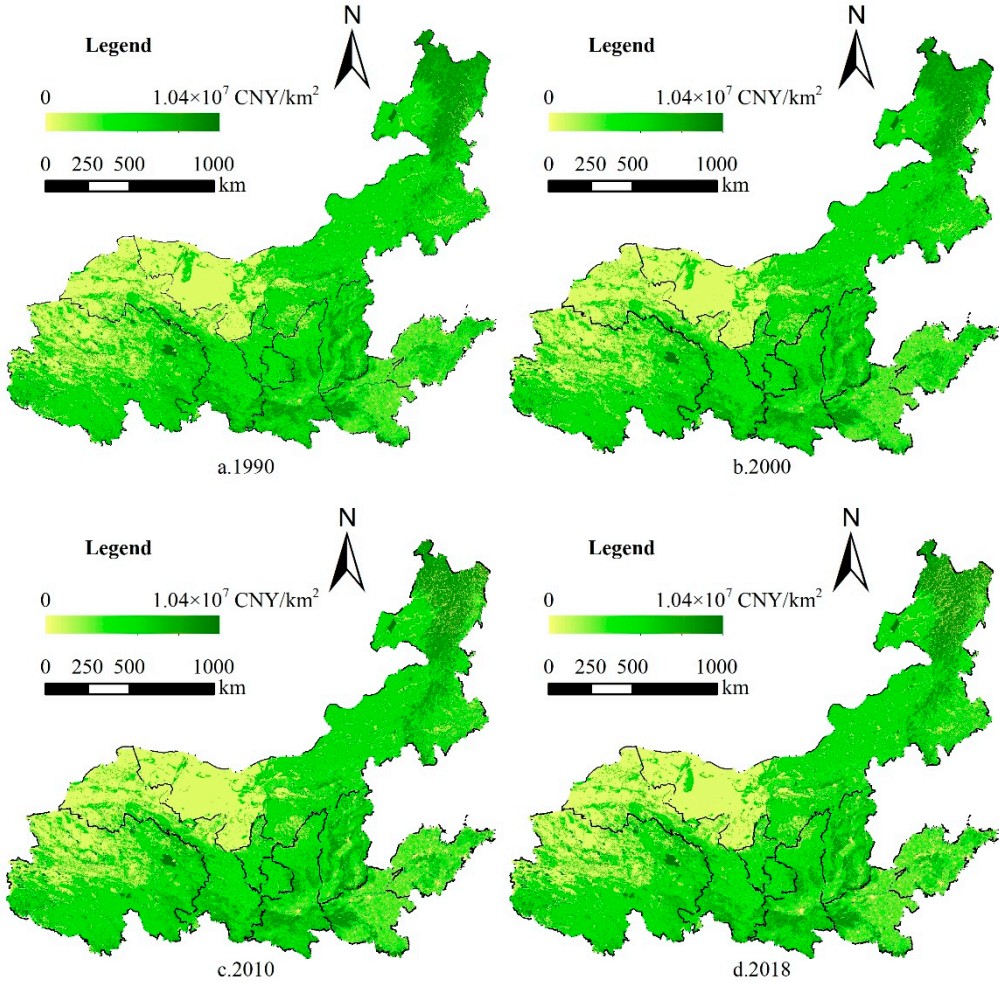

**Figure 4.** Spatial distribution of ESV per unit area in the YRB during 1990–2018.

The ESV remained stable in most areas of the YRB during the study period, with only a small number of areas improving or deteriorating (Figure 5). Specifically, the overall changes from 1990 to 2000 were minor, dominated by ESV deterioration, and were sporadic

across the region. From 2000 to 2010, nearly 10% of the regions changed in ESV, accounting for more than 70% of the regional changes during the whole study period. The Central Inner Mongolia Autonomous Region, and Shanxi and Henan Provinces were characterized by deterioration, while Qinghai and Shaanxi Provinces were characterized by improvement. The northeastern Inner Mongolia Autonomous Region showed mixed changes. From 2010 to 2018, ESV remained stable, with changes concentrated in south Gansu Province. In general, the ESV of YRB fluctuated during the study period, with an overall increase of 0.06%.

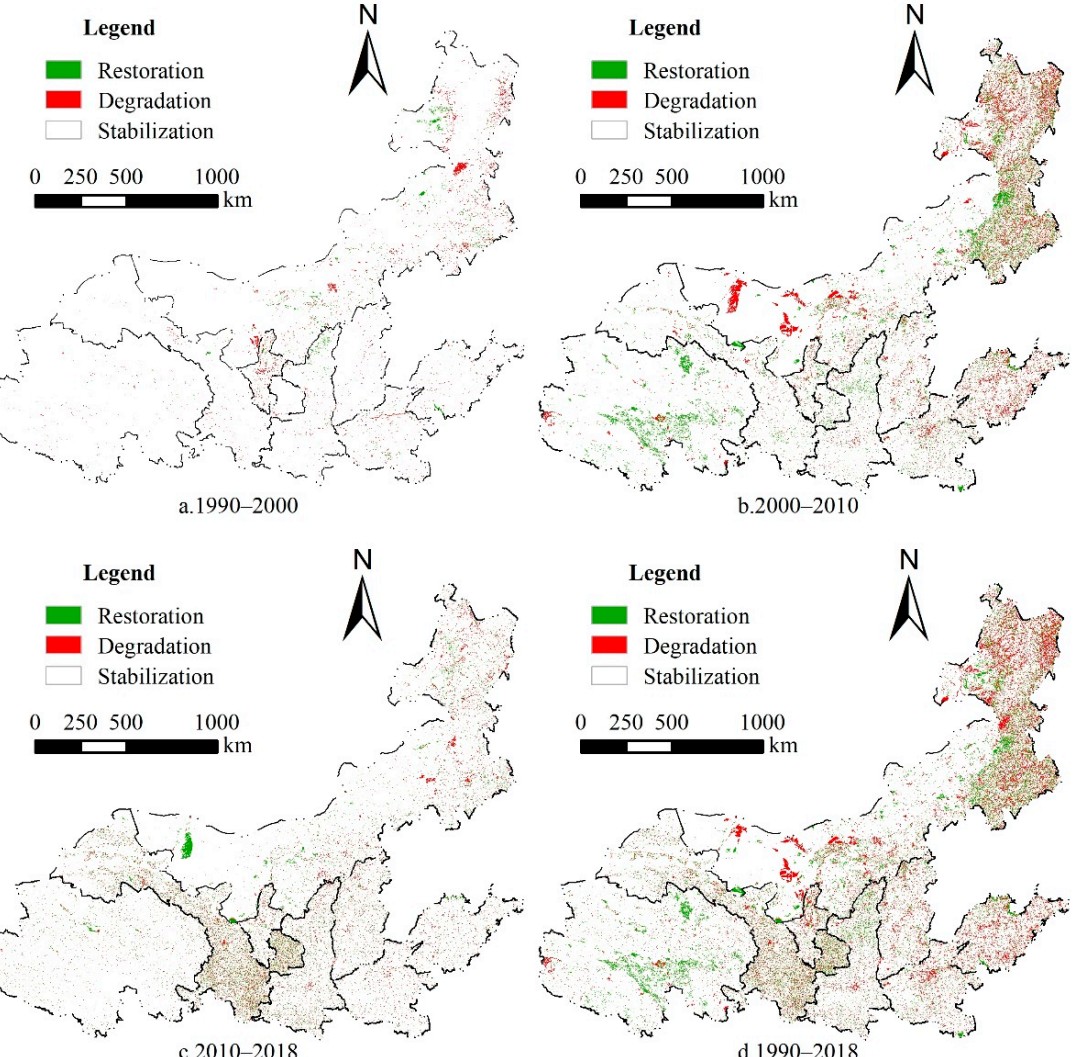

**Figure 5.** ESV changes in different study periods.

### 3.3. Impact of LUCC on ESV

#### 3.3.1. Ecological Contribution Rate of LUCC on ESV

ESV in the YRB was $680.03 \times 10^{10}$ CNY, $677.22 \times 10^{10}$ CNY, $678.91 \times 10^{10}$ CNY, and $680.45 \times 10^{10}$ CNY, respectively (Table 3). Grassland and forest land each contributed more than 40% and 30% of ESV, respectively. The contribution of forest land increased, while that of grassland decreased over time. The contribution of cropland remained consistent at around 13%, while the contribution of unused land was the lowest.

Different land use activities lead to different changes in ESV. Conversion from land use types with high value coefficients to those with low value coefficients will deteriorate ESV, while the opposite will improve ESV. According to the calculation results of Formula (6), a total of 30 pairs of land use type changes resulted in ESV variation (Table 4), of which half improved ESV and the other half deteriorated ESV.

**Table 3.** ESV coefficient of different land use type in the YRB (CNY·km$^{-2}$·a$^{-1}$).

| Type | 1990 | | 2000 | | 2010 | | 2018 | |
|---|---|---|---|---|---|---|---|---|
| | ESV | Proportion | ESV | Proportion | ESV | Proportion | ESV | Proportion |
| | 10$^8$ CNY | % | 10$^8$ CNY | % | 10$^8$ CNY | % | 10$^8$ CNY | % |
| Cropland | 8846.56 | 13.01 | 9053.06 | 13.37 | 8969.53 | 13.19 | 8865.89 | 13.03 |
| Forest land | 21,233.50 | 31.22 | 20,990.48 | 31.00 | 22,040.78 | 32.42 | 21,843.85 | 32.10 |
| Grassland | 29,731.38 | 43.72 | 29,460.94 | 43.50 | 28,383.45 | 41.75 | 28,453.66 | 41.82 |
| Water bodies | 6006.40 | 8.83 | 6035.81 | 8.91 | 6371.68 | 9.37 | 6689.31 | 9.83 |
| Unused land | 2185.46 | 3.21 | 2181.44 | 3.22 | 2215.17 | 3.26 | 2192.11 | 3.22 |
| Total | 68,003.31 | 100.00 | 67,721.73 | 100.00 | 67,980.62 | 100.00 | 68,044.81 | 100.00 |

**Table 4.** Contribution rate of land use change to ESV in the YRB during 1990–2018.

| Land Use Change | 1990–2000 | | | 2000–2010 | | | 2010–2018 | | | 1990–2018 | | |
|---|---|---|---|---|---|---|---|---|---|---|---|---|
| | Variation | Improvement | Deterioration | Variation | Improvement | Deterioration | Variation | Improvement | Deterioration | Variation | Improvement | Deterioration |
| | 10$^8$ CNY | % | % | 10$^8$ CNY | % | % | 10$^8$ CNY | % | % | 10$^8$ CNY | % | % |
| 1→2 | 28.26 | 5.71 | | 373.85 | 9.33 | | 183.93 | 8.05 | | 436.18 | 8.01 | |
| 1→3 | 34.33 | 6.93 | | 169.18 | 4.22 | | 184.19 | 8.06 | | 297.97 | 5.47 | |
| 1→4 | 56.82 | 11.47 | | 267.68 | 6.68 | | 199.53 | 8.73 | | 390.80 | 7.17 | |
| 1→5 | −54.68 | | 7.04 | −353.78 | | 9.44 | −192.99 | | 8.69 | −532.71 | | 9.85 |
| 1→6 | −6.21 | | 0.80 | −47.12 | | 1.26 | −27.89 | | 1.26 | −42.66 | | 0.79 |
| 2→1 | −124.08 | | 15.97 | −286.74 | | 7.65 | −198.63 | | 8.94 | −487.94 | | 9.02 |
| 2→3 | −128.05 | | 16.48 | −453.99 | | 12.11 | −463.44 | | 20.86 | −786.84 | | 14.55 |
| 2→4 | 1.51 | 0.30 | | 25.32 | 0.63 | | 12.75 | 0.56 | | 32.13 | 0.59 | |
| 2→5 | −2.69 | | 0.35 | −61.62 | | 1.64 | −46.86 | | 2.11 | −97.64 | | 1.81 |
| 2→6 | −5.68 | | 0.73 | −375.00 | | 10.01 | −173.83 | | 7.82 | −492.98 | | 9.12 |
| 3→1 | −128.39 | | 16.52 | −225.99 | | 6.03 | −182.79 | | 8.23 | −437.00 | | 8.08 |
| 3→2 | 62.91 | 12.70 | | 1209.61 | 30.19 | | 416.77 | 18.23 | | 1463.76 | 26.87 | |
| 3→4 | 46.70 | 9.43 | | 172.99 | 4.31 | | 183.12 | 8.01 | | 346.26 | 6.36 | |
| 3→5 | −7.34 | | 0.94 | −104.33 | | 2.78 | −109.24 | | 4.92 | −201.67 | | 3.73 |
| 3→6 | −110.00 | | 14.16 | −1176.80 | | 31.40 | −345.16 | | 15.54 | −1386.11 | | 25.64 |
| 4→1 | −112.94 | | 14.54 | −215.77 | | 5.76 | −122.51 | | 5.51 | −316.33 | | 5.85 |
| 4→2 | −2.15 | | 0.28 | −14.08 | | 0.38 | −10.37 | | 0.47 | −18.74 | | 0.35 |
| 4→3 | −31.99 | | 4.12 | −91.48 | | 2.44 | −139.78 | | 6.29 | −195.20 | | 3.61 |
| 4→5 | −3.02 | | 0.39 | −52.14 | | 1.39 | −49.64 | | 2.23 | −74.04 | | 1.37 |
| 4→6 | −59.21 | | 7.62 | −284.78 | | 7.60 | −153.69 | | 6.92 | −329.18 | | 6.09 |
| 5→1 | 0.59 | 0.12 | | 160.58 | 4.01 | | 74.53 | 3.26 | | 173.13 | 3.18 | |
| 5→2 | 0.23 | 0.05 | | 18.28 | 0.46 | | 14.47 | 0.63 | | 20.50 | 0.38 | |
| 5→3 | 0.00 | 0.00 | | 24.26 | 0.61 | | 31.24 | 1.37 | | 33.79 | 0.62 | |
| 5→4 | 0.10 | 0.02 | | 91.45 | 2.28 | | 99.90 | 4.37 | | 103.34 | 1.90 | |
| 5→6 | 0.00 | 0.00 | | 0.88 | 0.02 | | 1.47 | 0.06 | | 1.00 | 0.02 | |
| 6→1 | 19.01 | 3.84 | | 75.46 | 1.88 | | 49.69 | 2.17 | | 109.66 | 2.01 | |
| 6→2 | 8.74 | 1.76 | | 85.30 | 2.13 | | 115.59 | 5.06 | | 131.29 | 2.41 | |
| 6→3 | 97.97 | 19.78 | | 947.10 | 23.64 | | 454.35 | 19.88 | | 1251.04 | 22.96 | |
| 6→4 | 138.18 | 27.90 | | 385.04 | 9.61 | | 264.50 | 11.57 | | 657.45 | 12.07 | |
| 6→5 | −0.50 | | 0.06 | −4.35 | | 0.12 | −4.99 | | 0.22 | −7.75 | | 0.14 |

Note: 1–6 represent cropland, forest land, grassland, water bodies, built-up land and unused land, respectively. 1→2 represents land use type change from cropland to forest land, and other conversion types follow the same pattern.

The value coefficients of land use types determine the direction of ecological contribution of land use change, while the conversion area dominates the magnitude of contribution. Conversion from grassland to forest land and conversion from unused land to grassland during 1990–2018 were the key causes of ecosystem improvement, with their contribution rate more than 20%. Unused land converted into water bodies was a primary factor in ESV improvement, with a contribution rate of more than 10%. The conversion from cropland to forest land, grassland, and water bodies, as well as conversion from grassland to water bodies were minor factors for ESV improvement, with a contribution of more than 5%. Other land use change types contributed no more than 5% to ESV improvement and had only a negligible effect.

Conversion from grassland to unused land was the key cause of ESV deterioration, contributing more than 25%. Another primary factor for ESV degradation was the conversion of forest land to grassland, with a contribution rate of more than 10%. Furthermore, the occupation of cropland by expansion of built-up land, the conversion of forest land to grassland and unused land, the conversion of grassland to cropland, and the conversion of water bodies to cropland and unused land had less impact on ESV deterioration, with a contribution of more than 5%. Other land use change types contributed no more than 5% to the ESV deterioration.

3.3.2. Bivariate Spatial Autocorrelation between Land Use Degree and ESV

Using the GeoDa spatial analysis tool, a Queen spatial connectivity matrix was generated to calculate the global spatial autocorrelation index for land use degree and the value of each ESs in different years. As shown in Table 5, Moran's I for all ESs and land use degree was negative, except for supply services, indicating that there was a significant positive spatial correlation between supply services and degree of land use. This is because supply services are composed of food production and raw material, both of which are linked to the extent of cropland reclamation and built-up land expansion. As a result, increasing degree of land use results in improved supply services. Furthermore, there is a significant negative spatial correlation between land use degree and support services and cultural services, which are intimately linked to the natural ecosystem and its aesthetic landscape. There is no doubt that human efforts to strengthen land use have a negative influence on the natural environment, so the increase in degree of land use leads to a decrease in support and cultural services. The relationship between land use degree and regulation services was negative but not significant.

**Table 5.** Bivariate spatial autocorrelation between land use degree and ESV.

| Index | | Comprehensive Index of Land Use Degree | | |
|---|---|---|---|---|
| | | Moran's I | Z | *p*-Value |
| 1990 | Supply services | 0.4360 | 7.8729 | 0.0010 |
| | Regulation services | −0.0045 | −0.1379 | 0.4640 |
| | Support services | −0.1067 | −2.0931 | 0.0200 |
| | Cultural services | −0.3836 | −6.8991 | 0.0010 |
| 2000 | Supply services | 0.4299 | 7.7624 | 0.0010 |
| | Regulation services | −0.0121 | −0.2860 | 0.3870 |
| | Support services | −0.1106 | −2.1650 | 0.0150 |
| | Cultural services | −0.3825 | −6.8704 | 0.0010 |
| 2010 | Supply services | 0.3603 | 6.5781 | 0.0010 |
| | Regulation services | −0.046 | −0.9580 | 0.1790 |
| | Support services | −0.1821 | −3.5213 | 0.0010 |
| | Cultural services | −0.4051 | −7.2955 | 0.0010 |
| 2018 | Supply services | 0.3333 | 6.1278 | 0.0010 |
| | Regulation services | −0.0545 | −1.1116 | 0.1440 |
| | Support services | −0.2004 | −3.8674 | 0.0010 |
| | Cultural services | −0.4081 | −7.3436 | 0.0010 |

*3.4. Impact of Natural and Socio-Economic Factors on ESV*

3.4.1. Relative Effects and Interactions of Influencing Factors

Factor Detector results (Table 6) show that both natural and socio-economic factors affected the spatial heterogeneity of average ESV in the YRB, and the impact size of different factors changed slightly each year. NDVI had the greatest impact on the spatial heterogeneity of average ESV, with $q$ values of higher than 0.55 for each year. Meanwhile, the $q$ values of precipitation and population density were above 0.20, which were the primarily reasons for spatial heterogeneity of average ESV. In contrast, the $q$ values of slope, distance to road, distance to railway and distance to city were less than 0.10, and had smaller effects on the spatial heterogeneity of average ESV.

As revealed by the Interaction Detector (Table 7), the interaction effects between all pairs of factors selected were greater than those of each factor separately. As a result, the spatial heterogeneity of the average ESV in the YRB was caused by the mutual influence of multiple factors, and their interactions exacerbated the spatial heterogeneity. Specifically,

the interaction between NDVI and other factors had the strongest impact on average ESV. The $q$ values of NDVI $\cap$ population density were the highest ($q = 0.6605$), and thus had the strongest impact on the spatial heterogeneity of ESV. Following that were NDVI $\cap$ GDP ($q = 0.6564$) and NDVI $\cap$ elevation ($q = 0.6302$). There were 14 interaction combinations with $q$ values greater than 0.5, five were natural factor combinations, nine were natural and socio-economic factor combinations, with no combinations between socio-economic factors.

**Table 6.** The results of Factor Detector for the spatial heterogeneity of average ESV in the YRB during 1990–2018.

| Factors | 1990 | | 2000 | | 2010 | | 2018 | |
|---|---|---|---|---|---|---|---|---|
| | $q$ | $p$-Value | $q$ | $p$-Value | $q$ | $p$-Value | $q$ | $p$-Value |
| Elevation | 0.1140 | 0.0000 | 0.1105 | 0.0000 | 0.0998 | 0.0000 | 0.0961 | 0.0000 |
| Slope | 0.0717 | 0.0000 | 0.0717 | 0.0000 | 0.0825 | 0.0000 | 0.0829 | 0.0000 |
| Precipitation | 0.3861 | 0.0000 | 0.3873 | 0.0000 | 0.4066 | 0.0000 | 0.4031 | 0.0000 |
| NDVI | 0.5511 | 0.0000 | 0.5522 | 0.0000 | 0.5590 | 0.0000 | 0.5562 | 0.0000 |
| Population density | 0.2785 | 0.0000 | 0.2810 | 0.0000 | 0.2630 | 0.0000 | 0.2591 | 0.0000 |
| GDP | 0.1496 | 0.0000 | 0.1520 | 0.0000 | 0.1494 | 0.0000 | 0.1459 | 0.0000 |
| Distance to road | 0.0900 | 0.0000 | 0.0912 | 0.0000 | 0.0936 | 0.0000 | 0.0927 | 0.0000 |
| Distance to river | 0.1470 | 0.0000 | 0.1471 | 0.0000 | 0.1663 | 0.0000 | 0.1637 | 0.0000 |
| Distance to railway | 0.0183 | 0.0000 | 0.0188 | 0.0000 | 0.0231 | 0.0000 | 0.0228 | 0.0000 |
| Distance to city | 0.0448 | 0.0000 | 0.0454 | 0.0000 | 0.0472 | 0.0000 | 0.0475 | 0.0000 |

**Table 7.** The results of Interaction Detector for the spatial heterogeneity of average ESV in the YRB.

| $X_i \cap X_j$ | $q(X_i)$ | $q(X_j)$ | $q(X_i \cap X_j)$ | Interaction Types |
|---|---|---|---|---|
| $X_1 \cap X_3$ | 0.0961 | 0.4033 | 0.5650 | Nonlinear enhancement |
| $X_1 \cap X_4$ | 0.0961 | 0.5560 | 0.6302 | Two-factor enhancement |
| $X_1 \cap X_5$ | 0.0961 | 0.2592 | 0.6024 | Nonlinear enhancement |
| $X_1 \cap X_6$ | 0.0961 | 0.1459 | 0.5750 | Nonlinear enhancement |
| $X_2 \cap X_4$ | 0.0829 | 0.5560 | 0.5829 | Two-factor enhancement |
| $X_3 \cap X_4$ | 0.4033 | 0.5560 | 0.5952 | Two-factor enhancement |
| $X_3 \cap X_5$ | 0.4033 | 0.2592 | 0.5251 | Two-factor enhancement |
| $X_3 \cap X_6$ | 0.4033 | 0.1459 | 0.5017 | Two-factor enhancement |
| $X_4 \cap X_5$ | 0.5560 | 0.2592 | 0.6605 | Two-factor enhancement |
| $X_4 \cap X_6$ | 0.5560 | 0.1459 | 0.6564 | Two-factor enhancement |
| $X_4 \cap X_7$ | 0.5560 | 0.0929 | 0.5758 | Two-factor enhancement |
| $X_4 \cap X_8$ | 0.5560 | 0.1638 | 0.5810 | Two-factor enhancement |
| $X_4 \cap X_9$ | 0.5560 | 0.0228 | 0.5852 | Nonlinear enhancement |
| $X_4 \cap X_{10}$ | 0.5560 | 0.0475 | 0.6174 | Nonlinear enhancement |

Note: $X_1$—elevation, $X_2$—slope, $X_3$—precipitation, $X_4$—NDVI, $X_5$—population density, $X_6$—GDP, $X_7$—distance to road, $X_8$—distance to river, $X_9$—distance to railway, and $X_{10}$—distance to city.

### 3.4.2. Spatial Distribution of the Effects of Influencing Factors

Figure 6 depicts the spatial variation of the regression coefficients of each influence factor based on GWR results. The regression coefficients of NDVI are all greater than zero, meaning that NDVI is always positively correlated with ESV. Regression coefficients of GDP are generally greater than zero, indicating that the influence of GDP on ESV is mainly positive. In contrast, the regression coefficients of population density are generally less than zero, indicating that the influence of population density on ESV is mainly negative. In addition, other factors showed approximately equal areas of positive and negative correlation with ESV. Among them, the effects of elevation, precipitation, distance to road and distance to city on ESV were mainly positive in the west and negative in the east. The effects of slope and distance to railway on ESV showed mainly negative correlations in the west and positive correlations in the east, while the distance to river showed positive correlation in the center and negative correlations in the east and west.

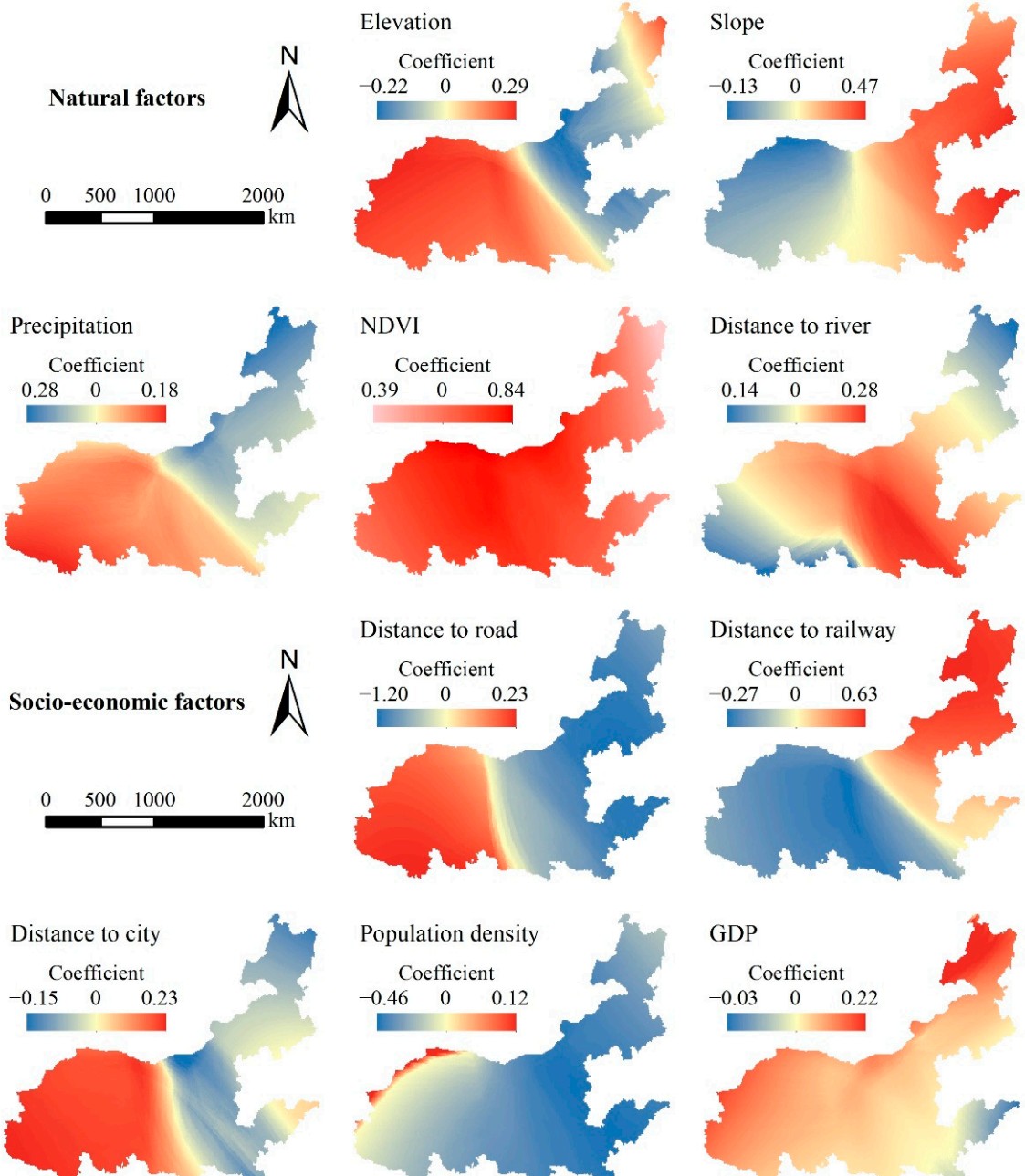

**Figure 6.** Spatial distribution of regression coefficients of influencing factors.

## 4. Discussion

### 4.1. Temporal-Spatial Evolution of ESV and Its Determinants

The findings of this study show that the northeastern part of the YRB with high vegetation cover had the highest average ESV, while the northwest part with scarce water resources and three large deserts had the lowest average ESV, which is consistent with the research of Cui et al. [43] and Zhang et al. [44]. Meanwhile, most scholars believe that the conservation of ecological lands such as forest land, grassland and water bodies is particularly important for the stability of ESV [34,35].

In terms of time, the ESV in the YRB clearly deteriorated from 1990 to 2000, owing to people's lack of awareness of the importance of their ecological environment and the failure to take effective ecological protection measures, resulting in the disorderly expansion of built-up land and the continuous degradation of forest land and grassland [45]. Since 2000, China has gradually strengthened ecological management, particularly through a

series of afforestation and soil conservation projects, which have improved the ecological environment of the YRB [46]. These ecological protection projects have shown preliminary results, indicating the state's important role in ecological regulation [47–49].

Overall, the total ESV of the YRB has remained stable, but the results show significant spatial heterogeneity, with some parts improving and others deteriorating. It was the balance of improvement and deterioration across the region that kept the ESV of the YRB relatively stable [50]. Regions with improved ESV were scattered across the YRB, all of them in areas showing an expansion of built-up land, and where at least one type of ecological land use had also increased. This suggests that the expansion of built-up land has been widespread, and that focusing on the cultivation of ecological land to allow orderly expansion of construction land can help to achieve the goal of ESV improvement [51,52]. Socio-economic development is based on the consumption of various resources, and land exploitation is inevitable. Our results show that any change in land use type resulted in a moderate or substantial change in ESV, exerting a direct impact on ESV. Specifically, the conversion of grassland to forest land was the major reason for ESV improvement in YRB, accounting for more than 25% of the total increase (Table 4), due to the much higher ESV per unit area of forest land than grassland (Table 2), and the large conversion areas (Figure 3). This was made possible by the implementation of projects such as the Three-North Shelterbelt Project, the "Grain-for-Green" and Natural Forest Protection programs, which resulted in a net increase in the area of forest land [53]. In particular, the Three-North Shelterbelt Project has been promoting large-scale afforestation since its implementation and has contributed greatly to ESV growth [54]. Meanwhile, the areas with deteriorating ESV were mainly concentrated in the eastern part of the YRB, and these areas showed a decline in forest land and grassland, and a significant increase in built-up land. Grassland degradation was the major factor in the deterioration of ESV in the YRB, accounting for more than 25% of the reduction (Table 4). Because of the fragile ecological environment of the YRB, with the frequent natural disasters such as floods and mudslides, grassland, a relatively ecologically fragile area, is vulnerable to destruction [33]. Additionally, human activities such as irrational use of water resources, overgrazing and overexploitation have exacerbated the degradation of grassland [55]. Therefore, in future construction, it is necessary to continue to supervise the implementation of these ecological projects, to strengthen the protection and construction of ecological land, and to formulate protection policies tailored to ecological degradation areas in order to ensure steady ecological improvement.

Research on ESV has become a hot topic in the process of building an ecological civilization. Most recent studies have found that changes in ESV are the result of a combination of natural and socio-economic factors [10,56]. Different factors can have varying impacts on ESV, and the combination of multiple factors can produce more complex effects [20]. From the perspective of sustainable development, positive impacts should be promoted and negative impacts should be suppressed [57]. Therefore, identifying the ways in which different factors contribute to ESV is essential for precise policy-making and the formulation of reasonable ecological regulatory measures. In this study, we used GDM and GWR models to explore in depth the impacts on ESV of multiple natural socioeconomic factors.

The results revealed that NDVI was the relative strongest influencing factor for spatial heterogeneity of ESV, which is consistent with the findings of Sun et al. [36]. Meanwhile, NDVI is the only factor that positively affected ESV in all regions. Because NDVI is an indicator of vegetation growth status [58], and abundant vegetation growth and cover are beneficial to ESV [59], therefore, NDVI had a relatively strong effect on ESV with a positive correlation across the whole area. Due to the different geographic conditions, there was spatial variability in the effects on ESV of all other factors. Overall, compared with socio-economic factors, natural factors dominated the influence on ESV. This is in line with the findings of Han et al., who suggested that various types of natural factors influence the structure, distribution, growth, and succession of biomes on a large scale, thus influencing ESV at a macroscopic level [49]. Nevertheless, the impact of socio-economic factors on ESV cannot be ignored, especially population density, which is the most influential of the

socio-economic factors. Population growth will lead to an increased demand for built-up land, food, and other necessities. In such a case, it will lead to LUCC and eventually influence ESV [51]. Therefore, in the practice of enhancing ESV, human interference should be minimized and the regulation of natural elements prioritized [60].

Furthermore, the results of the Interaction Detector revealed that the interaction effect of any two factors was greater than each factor alone, and the interaction type was dominated by two-factor enhancement, indicating that spatial heterogeneity of ESV in the YRB was the result of the combined effect of multiple factors. This is consistent with previous studies, which showed that combining various variables increased their influence on ESV [11,61,62]. It was noted that 9 of the 14 pairs of interaction combinations with $q$ values over 0.5 are interactions of nature and socio-economics, which means that in terms of interaction, the joint effect of nature and socio-economic factors has a stronger influence on ESV. Therefore, in ecological regulation, attention should be paid to the harmonious coexistence of man and nature, and to the combined effect of different regulation methods in order to maximize overall benefits and thus increase the efficiency of ecological improvement initiatives.

*4.2. Policy Implication*

Based on an in-depth analysis of the temporal-spatial evolution of ESV and its influencing factors, this study proposes three practical policy recommendations for the YRB. First, our study indicated that both in ecological improvement or deterioration areas, built-up land generally showed a trend for expansion, while the maintenance of ecological land such as forest land and water bodies can keep ESV stable. Therefore, an ecological monitoring mechanism can be established in the YRB to dynamically monitor various types of ecological land. The first priority is to monitor its area and limit the conversion of ecological land to other land uses. Ecological quality monitoring should also be enhanced for areas with ecological significance, and timely regulation should be carried out when their quality declines.

Second, the positive effect of the state as the main body to regulate ecology has already been shown, and the implementation of ecological conservation or restoration programs, such as the Three-North Shelterbelt Project, the "Grain-for-Green" Program and the Natural Forest Protection Program should be strengthened. Aside from policies applied to the entire region, specific programs should be designed and implemented in ecologically fragile areas based on their type of ecological vulnerability. For example, in water-scarce areas, a system of compensated use of water resources that matches socio-economic development should be implemented to conserve and control water in a comprehensive manner. In areas with severe land degradation, the local government should identify the type of land degradation, and implement unified planning and treatment in a piecemeal manner according to the classification results. In addition, areas prone to natural disasters should be designated as disaster management zones, requiring strict environmental control to prevent human activities from aggravating disasters.

Third, since natural ecosystems have the ability to self-heal, an ecological assessment mechanism can be established in the YRB, and different measures can be taken based on the assessment results to maintain or improve the ecological environment for different ecological zones. For example, in ecologically sound areas, ESV can be stabilized by limiting human activities, especially those that are polluting and destructive. In contrast, human interventions such as afforestation and engineering restoration are needed to rehabilitate the ecological environment in areas that have lost the ability to restore themselves.

*4.3. Limitation*

The YRB is an important ecological region within China. We investigated the temporal and spatial heterogeneity of its ESV, identified the ecologically fragile areas of the basin, evaluated the influencing factors, and proposed policy recommendations for improving the ecosystems of the YRB. Our findings may provide a basis for decision-making for ecological governance and regulation in the YRB.

However, there were some limitations to our study. First, we only classified the land use data into six primary categories without further subdividing it, which would tend to make the results biased. In the future, a more detailed classification of land use types should be performed to calculate ESV more accurately, such as dividing cropland into paddy fields, dryland, and irrigated land, and dividing forest land into tree forest, bamboo forest, shrub land, etc. Second, due to the limitation of data acquisition, the influencing factors used may not have been comprehensive enough, and policy or institutional factors were not taken into account. More data should be collected in the future to allow a more thorough investigation of the factors influencing ESV.

## 5. Conclusions

In this study, we investigated the temporal–spatial evolution of ESV and its determinants in the YRB, based on the ecological contribution, bivariate spatial autocorrelation, and geographical detector models. We found that the ESV of the YRB fluctuated during the study period, with an overall increase of 0.06%. Land use change exhibited a direct and dominant effect on ESV, with conversion of grassland to forest land and conversion of unused land to grassland being the dominant factors in ESV improvement, and conversion of grassland to unused land being the main cause of ESV deterioration. In addition, natural and socio-economic factors had a subtle influence on ecological elements, which gradually affected ESV. Furthermore, the differences in geographical location made the effect of natural socio-economic factors on ESV spatially heterogeneous. These results revealed that the adjustment of land use types in ecological management practices is a dominant factor in maintaining and improving ESV. At the same time, when formulating optimal land management policies, practical and efficient policies should be developed according to local conditions, to promote ESV improvement.

**Author Contributions:** Conceptualization, B.Z. and L.Z.; methodology, B.Z.; software, B.Z.; validation, B.Z., Y.W. and J.L.; formal analysis, B.Z.; investigation, B.Z. and L.Z.; resources, B.Z.; data curation, B.Z. and L.Z.; writing—original draft preparation, B.Z.; writing—review and editing, Y.W. and J.L.; visualization, B.Z.; supervision, J.L.; project administration, Y.W.; funding acquisition, J.L. All authors have read and agreed to the published version of the manuscript.

**Funding:** This research was funded by National Natural Science Foundation of China (Grant No. 41901213) and the Natural Science Foundation of Hubei Province (Grant No. 2020CFB856).

**Data Availability Statement:** The LUCC data were obtained from the Resources and Environmental Sciences and Data Center, Chinese Academy of Sciences (https://www.resdc.cn/ (accessed on 20 November 2021)). The administrative boundary data were obtained from the 1:400,000 database of the National Geomatics Center of China (http://www.ngcc.cn/ngcc/ (accessed on 22 November 2021)). The grain yield per unit area and grain price data were obtained Statistical Yearbooks of each province. Data for other influencing factors were obtained from the Resources and Environmental Sciences and Data Center, Chinese Academy of Sciences (https://www.resdc.cn/ (accessed on 20 November 2021)).

**Acknowledgments:** The authors are grateful to the editor and the learned reviewers for their valuable comments and suggestions.

**Conflicts of Interest:** The authors declare no conflict of interest.

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
