# Peer review of "Degradation or Restoration? The Temporal-Spatial Evolution of Ecosystem Services and Its Determinants in the Yellow River Basin, China"

_land, doi:10.3390/land11060863_

Round 1

Reviewer 1 Report

A very interesting article and this type of research is very much needed. However, due to the fact that the studied phenomena, eco-services are quite complex and the research is quite complicated, the most important issues should be selected from the research or the summary should be written in a less expert language. Thanks to this, the article would gain more readers from other scientific disciplines. I suggest authors to consider whether it would not be worth describing the conclusions in a less complicated language or adding a description of their own opinions. Each country has different specifics and the same changes can be assessed positively or negatively. A reader from another country, unfamiliar with this area, climate or the context of the discussed indicators, would prefer to find out clearly from the conclusions which changes (in the opinion of the research team) are assessed positively, and which are negative with justification. It is understandable only as a positive assessment of the conversion of wastelands into green and forest areas. The other changes are described in a somewhat complicated manner. Various indicators and relationships are cited and the reader may still be unsure whether or not they are assessed well as a result.

Reviewer 2 Report

The manuscript entitled "Degradation or restoration? The temporal-spatial evolution of ecosystem services and its determinants in the Yellow River Basin, China" use the contribution model to measure the contribution of intrinsic land use change to ESV, the bivariate spatial autocorrelation model was applied to investigate the relationship between land use degree and ESV, and the geographical detector model to detect the impact of natural and socio-economic factors on ESV, and then proposes relevant recommendations based on the findings. The article is clearly structured, the influencing factors are comprehensively considered, and the research methodology is informative. However, the following suggestions are made to the authors for improving their work:

1. The main problem of this paper is that the study area is not the Yellow River Basin, or it is inappropriate to take the provincial administrative division as the basin scope, which can not truly reflect the ecosystem service value and main factors of the basin. I suggest the author to recalculate and analyze the data after adjusting the scope of the study area. The scope of the Yellow River basin can be downloaded at https://www.resdc.cn/data.aspx?DATAID=141.

2. Line 113, it is suggested to make a detailed description of LUCC data, such as how many types of land use are divided and what is the classification accuracy?

3. Line 120, it is suggested to introduce the basic information of the selected natural factors and socio-economic factors in the form of a table, such as data source, selection year, data resolution, etc.

4. Fig.4, since you have reclassified the LUCC data by 1 km × 1 km, therefore, I suggest that the spatial distribution of average ESV in YRB should also displayed at 1 km resolution.

5. Please discuss in depth how the determinants affect the change of ESV?

Round 2

Reviewer 2 Report

The author has substantially revised the paper according to the comments, and I suggest accept in present form.